# Ranking Approach to Scheduling Repairs of a Water Distribution System for the Post-Disaster Response and Restoration Service

**Alicja Bałut** [1,*] , **Rafał Brodziak** [1] , **Jędrzej Bylka** [1] and **Przemysław Zakrzewski** [2]

1 Institute of Environmental Engineering, Poznan University of Technology, ul. Berdychowo 4, 61-131 Poznań, Poland

2 Institute of Computing Science, Poznan University of Technology, ul. Piotrowo 2, 60-965 Poznań, Poland

* Correspondence: alicja.balut@put.poznan.pl; Tel.: +48-61-665-2436

**Abstract:** On the maintenance task list of each water distribution system (WDS) operator, determination of the order of undertaken repairs seems quite a typical task. Characteristics of damages, their localization, and other factors that influence repair sequencing have a sound impact on the execution of such tasks. In the case of the most complex cases where numerous failures of different types occur at the very same time (i.e., due to earthquakes), there is a long list of selection criteria that have to be analyzed to deliver an objectively logical schedule for repair teams. In this article, authors attempt to find out if it is possible to define pipe rankings in having obtained the best factors for defined objective functions (criteria), making it feasible to deliver judicious repair sequencing. For the purposes of this paper, a survey has been carried out. Its conclusions made it possible to propose a method to create rankings of pipes and evaluate them using a selected multicriteria decision method: preference ranking organization method for enrichment evaluation (PROMETHEE). The work was carried out for five different disaster scenarios that had been supplied by 'The Battle of Post-Disaster Response and Restoration' organization committee. Obtained results might be further used to finetune this sequencing method of undertaken repairs, while conclusions could be useful to model similar events in WDS when required. This article is an extended paper based on the conference preprint presented at the 1st International Water Distribution Systems Analysis (WDSA)/International Computing & Control for the Water Industry (CCWI) Joint Conference in July 23–25, 2018 in Kingston, Ontario, Canada.

**Keywords:** hydraulic modelling; emergency response; PROMETHEE; multicriteria decision analysis

## 1. Introduction

Emergency management in water distribution systems is a subject that has been commonly addressed in publications over the last few years [1–4]. The problem that has been presented in this paper discusses a number of issues such as resilience of water distribution systems (WDSs) [5–9], valve topology for reliability analyses [10], optimization of restoration processes [11], design of water distribution systems [12–14], enhancement of strategies for resilience of water distribution systems [15], methodology of water networks modeling under emergency conditions [16], resilience index for urban water systems [17,18], and recovery strategy of water distribution systems after a disaster [13].

Resilience of WDS has been defined as the capacity to restore all operational functions and deliver safe drinking water in a fast manner after major disruptions. A range of reliability indicators and methods have been developed, each with a various degree of sophistication [19]. Most of them are based on the graph theory approach [9,20–22] and WDS modeling in uncertain circumstances. Still, there is a need for a more comprehensive indicator.

Security of water supply systems is related to two aspects. The first one has its roots in the need to provide a safe supply of water of adequate quality to end users. This can be implemented through appropriate process controls: water treatment, quality control of its intake, and produced water. The second aspect is related to the infrastructure of water supply networks and efficient maintenance of water transportation systems. Tasks that are related to it are: conservation and planning of networks' development, physical protection of sensitive points, and delivery of security policies (i.e., operating procedures that need to be undertaken) that need to be followed in special cases when, due to network damage, the supply of water has been breached. While single, accidental failures are usually handled according to the first in first out (FIFO) rule, as a deviation to this rule any damage to major network pipes would be handled with higher priority.

While the approach described above can be applied to the "daily" situations, it is not sufficient in handling failures in cases of emergency—i.e., earthquakes or terrorist attacks—when numerous damages occur simultaneously and in various places across the WDS. Restoration of a water distribution system that has been damaged by an earthquake was the subject participants were asked to analyze at the eighth battle competition of the Water Distribution Systems Analysis conference—The Battle of Post-Disaster Response and Restoration (BPDRR). This BPDRR problem highlighted some of the decisions that would have to be made by any water utility once affected by a disaster. Its purpose was to explore how available resources should be best utilized for restoration needs and also to identify how water utilities should get prepared for such scenarios. The main goal was to present the method to schedule operations of pipe repairs and repair works in order to restore normal working conditions of the WDS.

The main assumptions of BPDRR are presented below, and a detailed description of the problem is included in [23]. Organizers delivered a calibrated Water Distribution Network (WDN) model of the B-City and five sample scenarios of damages likely to occur during an earthquake, whose impact range was estimated on the city's seismic conditions. Each scenario contained a list of pipe damages—breaks and leaks. Repair methods were individually assigned to each type of damage. It was then the participants' decision to decide how to schedule the available three crews to isolate, repair, or replace damaged pipes. Crews could perform repairs in parallel while working on various damages. Depending on their types, restoration time and chosen methods were, of course, different. Moreover, restoration of critical network nodes—i.e., hospitals or firefighting flows—had to be taken care of with higher priority. Additionally, each damage was further categorized as visible or invisible, depending on the time accidents happened and diameters of damaged pipes.

This paper describes and analyses a few proposed strategies on how to respond to emergencies, with focus set on how to prioritize usage of its available resources to restore services. Planning of remediation actions requires correct information (i.e., hydraulic models of water supply networks or accurate details of damages that occurred). In order to compare the effectiveness of the proposed solutions in restoring the water supply networks, each repair scenario had to be assessed using defined objective functions. The evaluation criteria had been predefined by BPDRR organizers:

- C_01—length of time hospitals and firefighting flows were without supply,
- C_02—rapidity of recovery (length of time until the system's functionalities were permanently recovered by 95%),
- C_03—resilience loss (resilience in percent of water lost over seven days that followed the event),
- C_04—average length that time nodes were without service, in minutes,
- C_05—number of nodes that awaited servicing for more than 8 consecutive hours,
- C_06—volume of water lost over seven days that followed the event.

To gain real insight into how such a problem is approached across the industry, we sent a questionnaire to decision makers of water utilities based in averaged sized municipalities across Poland. Responders were asked to list the main criteria that in their opinion influenced the sequence of repair scheduling. Also, they were asked to share their feedback on what other factors impacted scheduling

in case of an earthquake or a similar event, if it happened. It turned out that in case of natural disasters, simple solutions were mostly sought out to set rules of conduct. They insisted that the method had to be based on expert knowledge backed up by IT solutions, and that it should be universally applicable, irrespective of any given scenario or network type.

## 2. Methodology

The aim of the work was to provide an answer to the question that had been stated: 'Is it possible to determine the prioritization of water pipes in order to define the sequence of repairs?' All undertaken steps were grouped into stages; they are presented in Figure 1, which presents the main steps and tools used in method delivery. The starting point for analyses was a review of feedback received to the questionnaire that had been earlier sent out. Once done, hydraulic and GIS (Geographic Information Systems) analyses of the network were conducted. Conclusions drawn upon questionnaires and hydraulic analyses were then applied to rank the importance of the pipes, which was then further used to sequence pipe repairs.

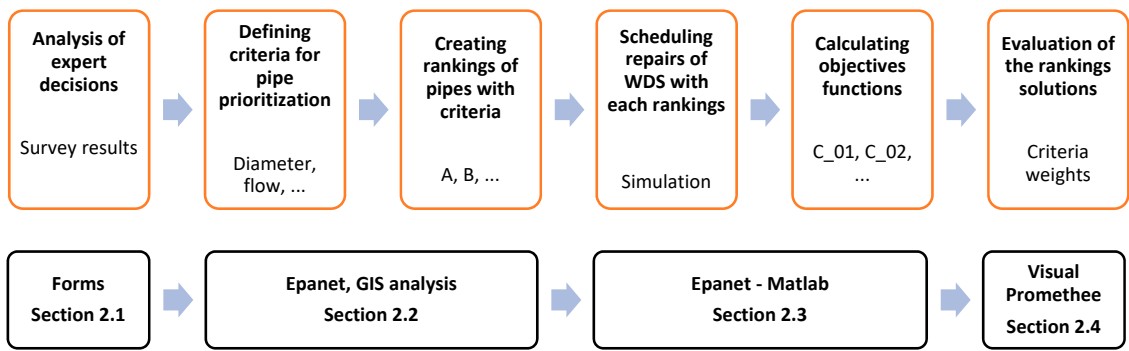

**Figure 1.** The upper diagram presents the flowchart of the proposed methods. The lower one—matching tools per each step.

### 2.1. Survey Results

The questionnaire that was conducted surveyed responders on multiple breakdowns in the water network and on their decisions. Forty-six top management employees from water utilities participated including CEOs, CTOs, and also consultants, IT specialists, and water distribution system modelers. The survey included a diagram of a water supply network (Figure 2) that was an excerpt of the BPDRR model. Breakdowns were marked in red, leaks in black, and critical points (hospitals, fire-flow nodes) were highlighted.

Questions concerned the determination of the sequence order repair tasks were to be completed on that water network. The key focus of this survey was to understand how, in such cases, real life decisions are made. There were two assumptions we decided to make:

- There were 8 breakdowns,
- There were 2 repair crews available.

Each repair crew was assigned 4 tasks. The answer provided most frequently has been bolded. Table 1 presents a matrix of responses: breakdown IDs versus the sequence in which they should be overhauled, with a focus on 4 initial tasks.

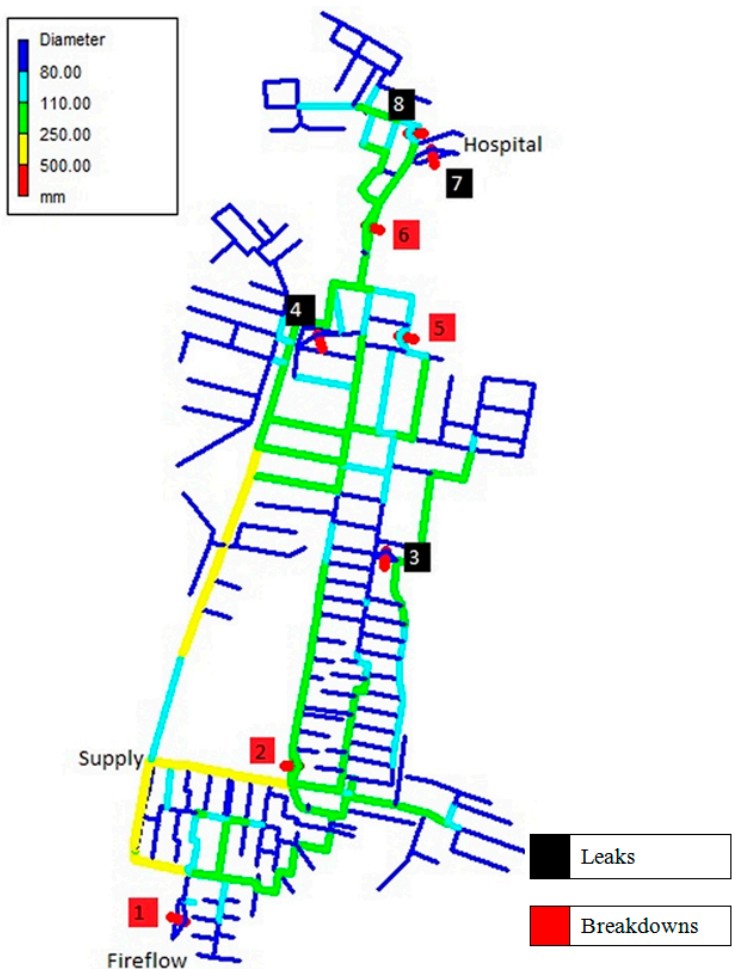

**Figure 2.** Water supply network diagram from the survey with marked breakdowns.

**Table 1.** Survey results—number of answers per each breakdown repair ID.

| Team | Breakdown ID | Number of Answers | | | |
|------|--------------|--------|--------|--------|--------|
| | | Task 1 | Task 2 | Task 3 | Task 4 |
| 1 | 1 | **19** | 3 | 0 | 2 |
| | 2 | 4 | **18** | 1 | 0 |
| | 3 | 1 | 0 | **13** | 9 |
| | 4 | 0 | 8 | 5 | **10** |
| 2 | 5 | 0 | **10** | 8 | 5 |
| | 6 | **18** | 1 | 4 | 0 |
| | 7 | 1 | 3 | **11** | 8 |
| | 8 | 3 | 3 | 4 | **12** |

Survey results were used to determine the order of tasks assigned to each of the crews. Table 2 presents the determined order of task delivery.

**Table 2.** The results of survey—order of task delivery.

| Task | Breakdown ID | |
| :---: | :---: | :---: |
| | Crew 1 | Crew 2 |
| 1 | 1 | 6 |
| 2 | 2 | 5 |
| 3 | 3 | 7 |
| 4 | 4 | 8 |

The following rough approach could be drawn upon the analysis of responses to the questionnaire: breaks on the way to strategic points should be repaired first, followed by breaks on the pipes with a larger diameter, and then all other breaks closing the list. Additionally, responders were asked to assign weights (from 0 to 10) to 4 defined criteria that addressed the rapidity of recovery, number of nodes without service, and volume of water lost. Such an approach derived from the need to simplify the questionnaire and to acquire potential preferences in the decision-making process.

It was not the purpose of this paper to publish questionnaire results since, due to the limited number of responses received, it did not have adequate statistical power. Instead, the intent behind the questionnaire was to receive some 'voice of the decision makers'. Survey results were applied as the direction in the decision-making process, not as the decision made arbitrarily.

## 2.2. Developing the Ranking

The research that we undertook was conducted on a hydraulic model provided by BPDRR organizers [23]. It contained 4909 junctions, 6064 pipes (402 km), 1 reservoir, 4 pumps, and 5 tanks.

During the first hours of the simulation, in some nodes there were firefighting demands—35 L/s (please check 'Damage_Scenarios.xlsx' attached into Supplementary Materials). As dictated, each site on fire was extinguished with precisely 756 m$^3$ of water. However, since pressure-deficient conditions were met in the network during most of the restoration time, the demand could not be fully supplied. To model partial supply, a pressure-driven analysis (PDA) must be used.

Water supply ($Qi$) in node *i* is, therefore, the function of the pressure head ($pi$), the demand in node *i* ($Q_{Di}$), and the pressure head required ($p_{req}$) to fulfill that demand in node *i* ($p_{req} = 20$ m). Results of each of five scenarios were evaluated using provided EPANET2 [24] models. In order to solve the PDA model of a WDN with regards to the flow–pressure relationship, Wagner's power relationship (1) was used.

$$Q = \begin{cases} 0, & p < 0 \\ Q_D \left( \frac{p}{p_{req}} \right)^{0.5}, & 0 < p \leq p_{req} \\ Q_D, & p > p_{req} \end{cases} . \tag{1}$$

To solve this equation, four types of artificial Epanet elements: Flow Control Valve (FCV), Throttle Control Valve (TCV), Check Valve (CV) and reservoir (R) were added to all demand nodes in the model, allowing this algorithm to enforce pressure-driven demand. This noniterative method allowed us to calculate the available flow during failure conditions. Further details on PDA calculations in Epanet can be found in article [25]. Various failure scenarios were analysed using this model as the basis. The main goal of this task was to determine the work schedule for 3 teams that could repair different breakdowns in parallel.

The detailed description of the problem along with required tasks are presented in the 'Problem description' in [23]. Having the target to find the solution to the BPDRR problem, hydraulic analyses on the obtained base model were performed.

Analyses were conducted on that water network, and flows were read at 9:00 AM—the peak time of maximum demand. The results are presented in this paper in Figures 3 and 4. Figure 3 focuses on the distribution of diameters, and Figure 4 focuses on hydraulic conditions and the distribution of flow.

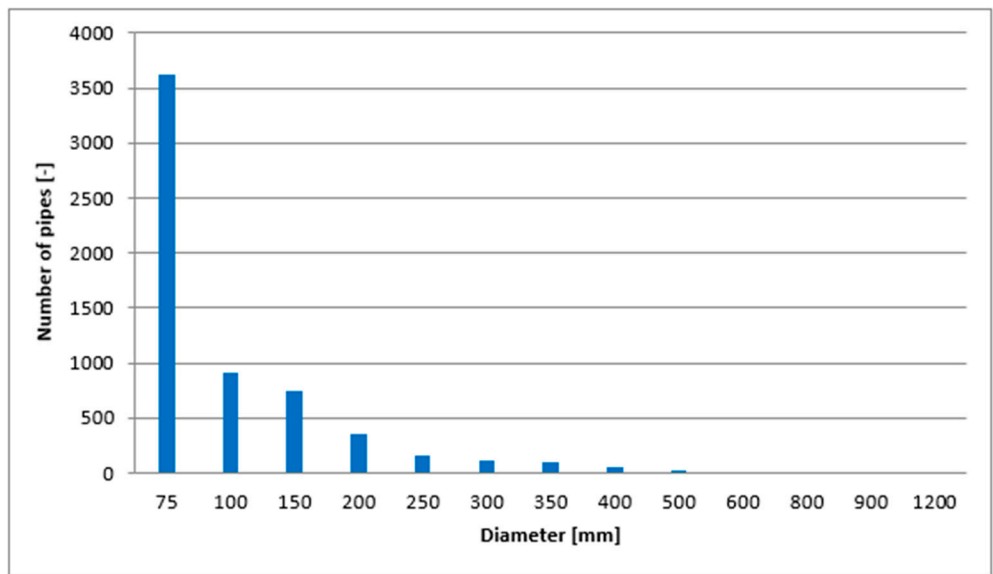

**Figure 3.** Diameter of pipes in the network.

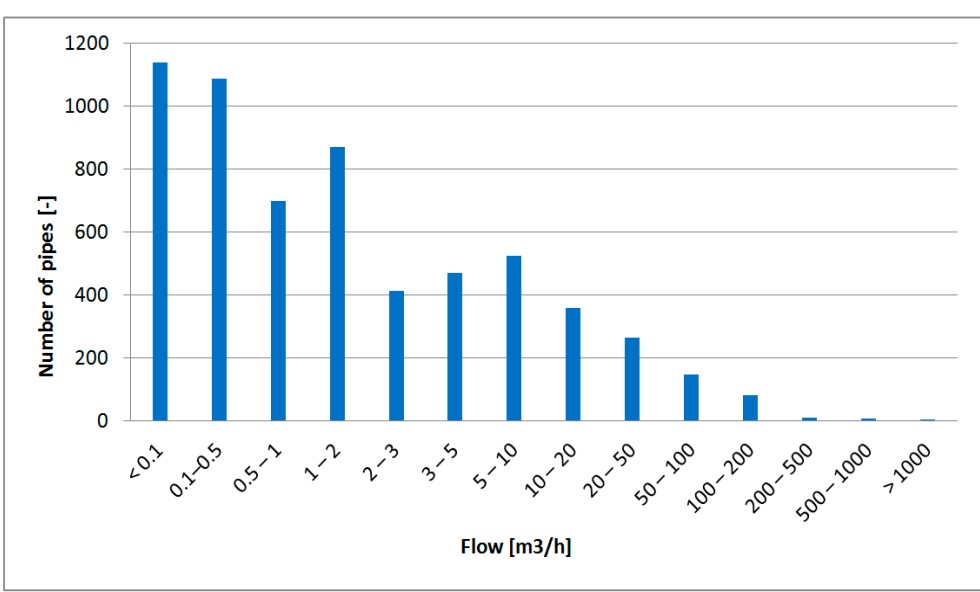

**Figure 4.** Flows in the pipes in the network.

An analysis of the network's hydraulics may lead to the assumption that, due to a high value of diameter/flow, only a limited number of pipe sections are hydraulically important, whereas the significance of the other ones is similar. Our approach to finding the solution to the stated problem assumed that first a list of water pipes should be created and then each classified with various parameters (i.e., A—diameter, B—diameter and distance from source, C—diameter and velocity, etc.).

The setup of the prioritization of rankings had to fulfill the following assumptions:

- Rankings had to be developed on the provided water distribution network model that had been built under normal working conditions (BPM-EPS).
- Rankings had to be created as a part of a strategic document (i.e., the water safety plan). The prioritization scheme had to address all types of possible accidents (earthquakes, floods, wars, etc.) that could cause numerous breaks in water networks. All elements of water distribution networks had to be taken into account (water intake, water treatment stations, and water distribution networks).

- Development of the ranking had to be 'as easy as only possible'. The key issue water utilities face, especially those from Central and Eastern Europe, is a lack of detailed data related to their networks. Only selected utilities possess reliable data on their networks and also calibrated models. Thus, the root cause behind the development of any ranking is that, in case of any damage, the strategy for network operators to follow had to be clear and intuitive. Events as extreme as earthquakes give no time to compute various scenarios, search for lacking data, research for operational hydrants, and so on. In our opinion, such events required 'ready to do' lists set up beforehand that covered the most critical pipes that, in turn, determined the sequence of next actions to be performed.
- During development of the rankings it was possible to include the application of compensational tanks, yet only under the condition that the volume of water collected for emergency purposes had been clearly defined.

The proposed method of ranking approach to scheduling repairs of water distribution system consisted of three steps: The goal of the first one was to select prioritization criteria and construct the ranking. The purpose of the second step was to calculate values for all criteria that had been provided in the problem descriptions to evaluate the performance of the target solution, while the third step focused on choosing the best ranking for the scenario. These are the criteria we proposed to set up rankings for the given problem:

A—Diameter

This ranking is based on the pipe diameter only. Figure 5 presents the choice of five main pipes with largest diameters that should be repaired first, followed by the others with smaller diameters (Figure 6).

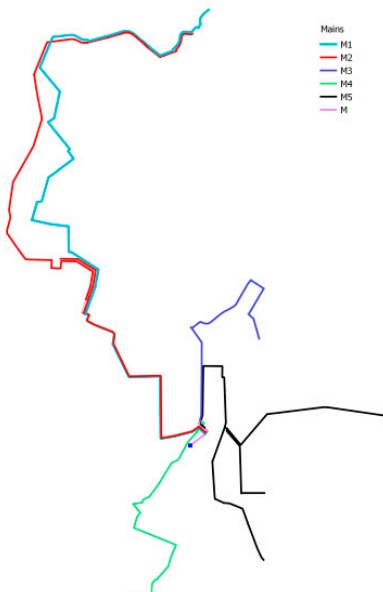

**Figure 5.** Map of the water supply network—main water pipes.

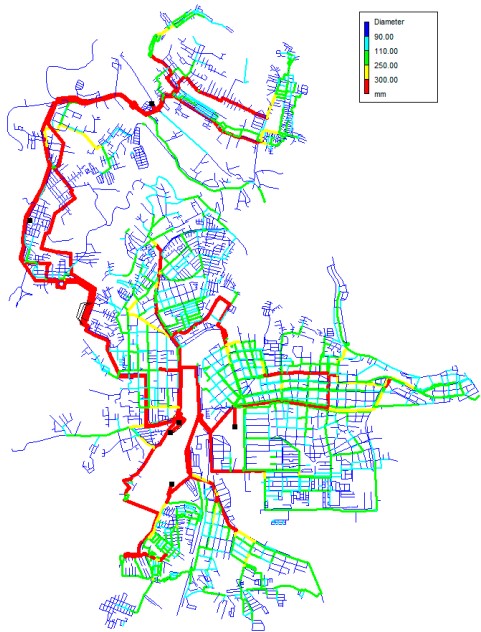

**Figure 6.** Map of the water supply network—diameter of pipes.

Rankings based on the 'diameter size methodology' gives maintenance priority to pipes with the largest diameters. The main advantage of this method is its simplicity, and that such a ranking can be developed without a calibrated model of a given water distribution system. The main disadvantage, however, is that it does not take into account hydraulics nor topology of the network (flows, velocity). It is also not possible to distinguish the importance of pipes of the same diameter (the search for pipes with the same diameter randomizes their position in the ranking).

B—Diameter and distance from source

According to this methodology, two criteria can be taken into account: pipe diameters and distance between their counterpoints and the source. This simple method does not require a calibrated model of a water network, yet it partially includes hydraulic characteristics of the network and its topology.

C—Diameter and velocity

This methodology sets the pipe diameter as the key criterion. Prioritization is calculated based on the hydraulic load of pipes. It is assumed that for pipes of the same diameter, higher velocity shifts their priority in the ranking.

D—Flow

This ranking methodology sets priority to pipe sections with the higher flow at the time of maximum demand (Figure 7). Network hydraulics are taken into account.

E—Flow (including strategic points)

The logic behind the D ranking makes use of the flow in the network—pipes with higher flows (at the time of maximum demand) are set higher on the repair sequence list. The E ranking additionally includes the priority of strategic points (for hospital and fire flow nodes, an additional flow of 35 liters per second is added). Due to negative pressure, such a model does not correspond to reality, but it allows us to determine a hydraulic path to the strategic points. Using this method, it is possible to include all types of strategic points. Figure 8 presents the distribution of water fire flow where this type of ranking is based.

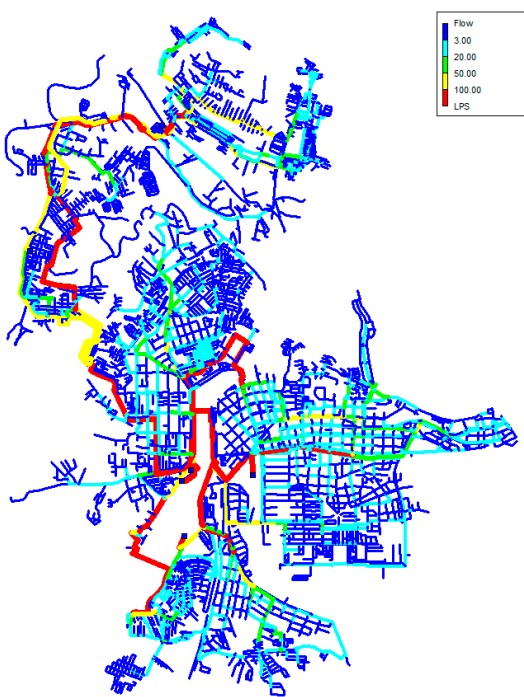

**Figure 7.** Map of the water supply network—flows in pipes: base model.

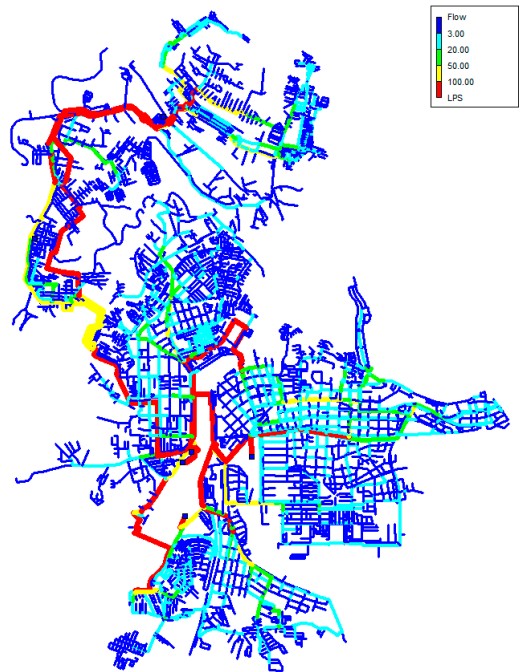

**Figure 8.** Map of the water supply network—flows in pipes: base model with fire flow nodes.

F—Impact of pipe closures on network hydraulics

This approach is based on an assumption that in every network there are pipe segments where any break causes a pressure drop in the whole network. In the analysed network, closure of pipe #168 caused a negative pressure across the entire network (Figure 9), while closure of pipe #88 only resulted in a local drop of pressure (Figure 10).

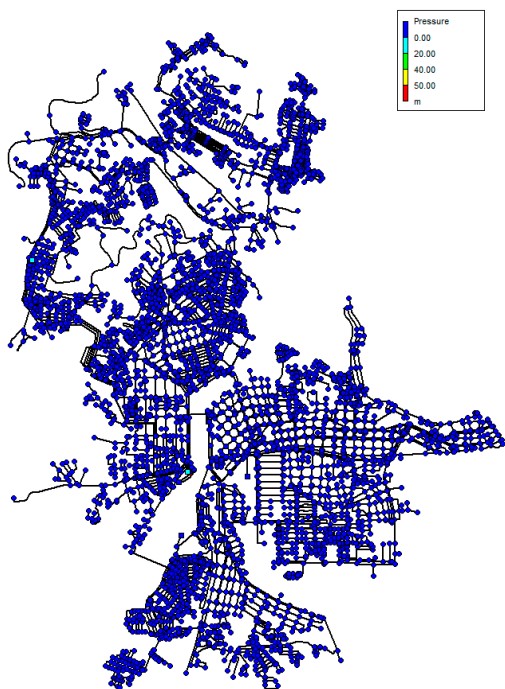

**Figure 9.** Map of the water supply network—pressures after closure of pipe #168.

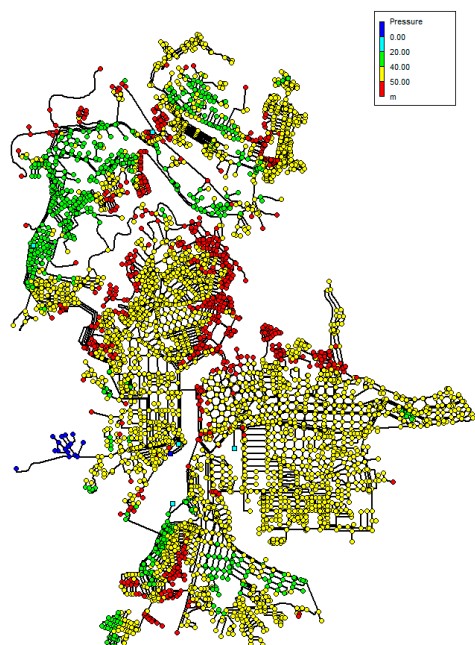

**Figure 10.** Map of the water supply network—pressures after closure of pipe #88.

During normal system exploitation, the water supply to every node and in all time steps was met. Thus, closure of a single pipe results in decreased pressure. In the course of analysis for certain nodes, negative pressure was received. This is interpreted as a lack of water supply to consumers that have been assigned specific nodes. Consumers with no access to water supply (calculated as the sum of water usage in out of service nodes) is the measure of the influence of a pipe's closure on network hydraulics.

*2.3. Calculation Methods*

The Figure 11 presents the general calculation flowchart for the delivery of the repair schedule.

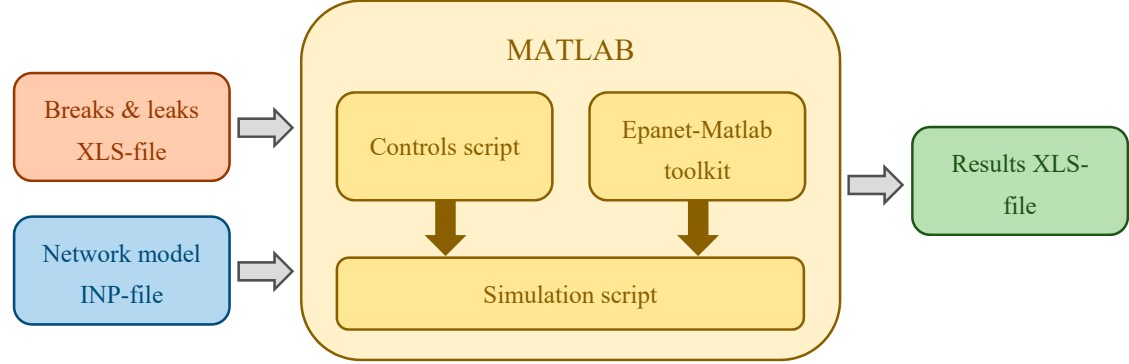

**Figure 11.** General calculation flowchart for delivery of the repair schedule.

Input data upon which calculations were to be based were provided via a file that contained the hydraulic model of the damage scenario and the list of breaks and leaks. Simulations were conducted using the Matlab environment. Epanet's model calculation was executed directly in Matlab with the Epanet–Matlab toolkit installed [26]. This package utilizes the Epanet.dll library for hydraulic calculations, which enables further processing of objective functions. Setup of controls within the hydraulic model was required to model the maintenance of brakes and leaks (isolation, reparation, and replacement). A dedicated Matlab script was prepared solely for that purpose. Calculation of the objective functions for 6 criteria required around 1 h. It is worth mentioning that the complete calculation process was realized on a standard PC (Intel i5, 16Gb) with average processing speed. If any real deployment were to be found for solutions of this type, it would be required to specify the time necessary for an operator to make a decision with the use of our solution as the decision support system.

Our solution's purpose was to generate the sequence of tasks that in the following steps needed to be assigned to individual repair teams. A separate list of tasks was then generated for each pipe damage. Table 3 presents such a sample tasks list.

**Table 3.** Example list of tasks for particular damages.

| Break/Leak | Diameter | Pipes to Isolate for Repair Break | Notes | Valves | Closing Time | Average Time |
|---|---|---|---|---|---|---|
| ID | (mm) | ID (*) | (-) | (-) | (min) | (h) |
| 437 | 500 | 437 (0), 599 (1), 6069 (1), 169 (1) | | 3 | 45 | 14.00 |
| 3602 | 200 | 3602 (2) | | 2 | 30 | 7.00 |
| 2112 | 200 | 2112 (1), 2106 (1), 2123 (1), 2124 (1) | | 4 | 60 | 8.00 |
| 3562 | 200 | 3562 (1), 3584 (1), 3721 (1), 3720 (1) | important | 4 | 60 | 8.00 |

* how many valves belong to the pipe.

During the next stage, the list of tasks to be repaired was sorted by pipe 'importance', in accordance with the selected ranking. The team that became available was then assigned a sequence of repair works. This process was complete once all teams received their task lists. These, initially as text files with controls, were then imported into the Epanet model. During simulation of the repair schedule model, objective functions were calculated, and results were saved as a file. To sum up, for all 6 proposed rankings and 5 different sample scenarios, repair works were scheduled and simulated for all teams in accordance with the method that has been presented above.

*2.4. Evaluation of the Ranking Solutions*

In order to find out which of the rankings appeared to best answer all given scenarios, Visual PROMETHEE was used to evaluate them, based on weights assigned to criteria [27]. PROMETHEE I and II are relatively simple ranking methods that were initially delivered by Brans in 1982 [28], with latest VI generation on the market today. There are several specialized software programs, including

PROMCALC, DECISION LAB 2000, Visual PROMETHEE, and D-Sight [29], where PROMETHEE methods have been applied and delivered to the market as tools that could be used to provide needed solutions and to decrease the decision-making time.

The choice of Visual PROMETHEE was made because of its key ability to handle differences among evaluations of variants, made for all criteria. Multicriteria decision analysis (MCDA) helps decision makers incorporate various, often conflicting, priorities into an evaluation of alternatives. The best solution is selected out of all alternatives in the process of performance assessment that is based on preselected criteria [30–33].

In the PROMEHTEE II method, the search for the optimal solution has been included in five implementation steps:

- Step 1—designation of preference function values for all variants and all criteria. This function needs to be separately defined for each criterion, and each needs to receive a value from the range between 0 and 1. The smaller the function the greater the indifference of the decisionmaker; the closer to 1 the stronger their preference.
- Step 2—designation of equivalence thresholds for all variants and all criteria.
- Step 3—designation of preference thresholds for all variants, individually for each criterion.
- Step 4—determination of the multicriteria preference index and, finally, the matrix of indicators, based on the preference degree:

$$\pi(a,b) = \frac{1}{k} \sum_{h=1}^{k} P_h(a,b), \tag{2}$$

where $k$ represents the total number of criteria, $h = 1, 2, \ldots, k$, $a, b \in K$ is the quantity of alternatives, and $P_h(a, b)$ expresses the preference function between $a$ and $b$ alternatives.

- Step 5—settlement of the ranking of variants based on net dominance flows, defined as:

$$\Phi^+(a) = \sum_{x \epsilon K} \pi(a,x), \tag{3}$$

$$\Phi^-(a) = \sum_{x \epsilon K} \pi(x,a). \tag{4}$$

A positive net flow (outgoing flow) expresses the degree to which the considered variant outmatches all the other ones—Equation (3). Negative net flow (incoming flow) expresses the degree to which the considered variant is outranked by all others—Equation (4). For this BPDRR case it is the PROMETHEE that was chosen, mainly because the analysed alternatives (Section 2.2 Developing rankings) were not comparable in this method. To apply the PROMETHEE method, Visual Promethee Academic Edition [34] was selected to rank all six types of alternatives (Section 2.2).

The following are indispensable to sequence available alternatives by Visual Promethee:

(a) Preference measure (evaluation matrix),
(b) Weights,
(c) Preference function.

Ad. a) Preference measure (evaluation matrix)

Input evaluation matrix for PROMETHEE is the result of calculations described in Section 2.3 'Calculation Method'. Table 4 presents the formulated evaluation matrix. This matrix lists alternatives ranking from A (diameter), B (diameter and distance from the source), C (diameter and velocity), D (flow), E (flow including strategic points), and F (impact of pipe closures on network hydraulics).

Results that best met the required criteria are bolded and framed. In the green background, however, these alternatives were listed that met the highest number of criteria at a time, under the condition that all weights are equal. Having reviewed feedback received from water utility staff,

it turns out that decision makers subjectively applied different weights to six criteria already discussed. Objectively, alternatives marked green are not optimal in all cases. Following this input, the influence of applied weights on achieved results has had to be considered (Table 6).

**Table 4.** Input evaluation matrix for the preference ranking organization method for enrichment evaluation (PROMETHEE).

| Scenario | Alternatives | Results from MATLAB and EPANET Matlab Toolkit | | | | | |
|---|---|---|---|---|---|---|---|
| | | C_01 | C_02 | C_03 | C_04 | C_05 | C_06 |
| | | (min) | (min) | (min%) | (min) | (-) | (L) |
| Scenario 1 | A | 3000 | **460** | 41,648 | 216 | 801 | 76,913,854 |
| | B | 3165 | **460** | 42,333 | 213 | 795 | **77,166,926** |
| | C | 3045 | **460** | 41,324 | 212 | **759** | 77,640,589 |
| | D | **2970** | 460 | **41,098** | **211** | 760 | 77,370,499 |
| | E | **2970** | 460 | 42,051 | 217 | 793 | 78,046,971 |
| | F | 3555 | **460** | 47,564 | 261 | 799 | 85,936,068 |
| Scenario 2 | A | **3510** | 276 | 19,697 | 46 | 134 | 57,045,249 |
| | B | 3765 | **276** | 20,129 | 48 | 143 | 56,725,233 |
| | C | 3720 | **276** | 19,233 | 44 | 135 | 56,820,145 |
| | D | 3795 | **276** | 19,379 | 45 | 138 | 56,480,788 |
| | E | 3720 | **276** | 19,182 | 42 | 134 | 56,988,294 |
| | F | 3960 | **276** | **18,608** | **37** | **120** | **59,438,741** |
| Scenario 3 | A | 3690 | **364** | **26,224** | 36 | **73** | 82,986,155 |
| | B | 3720 | **364** | 28,692 | 38 | 78 | **82,561,084** |
| | C | 3690 | **364** | 28,597 | 38 | 77 | 84,812,665 |
| | D | **3375** | **364** | 28,207 | 38 | 76 | 82,910,628 |
| | E | 3480 | **364** | 28,927 | 38 | 74 | 84,615,279 |
| | F | 3900 | 457 | 29,308 | **34** | 79 | 91,362,983 |
| Scenario 4 | A | **2565** | **364** | **32,555** | **80** | **166** | 74,506,055 |
| | B | 2430 | **364** | 33,823 | 82 | 198 | 74,246,491 |
| | C | 2610 | **364** | 33,063 | **80** | 168 | 75,216,764 |
| | D | 2445 | **364** | 33,470 | 81 | 168 | **75,133,769** |
| | E | 2625 | **364** | 32,949 | 81 | 171 | 75,596,054 |
| | F | 2790 | **364** | 36,002 | 103 | 399 | 78,722,686 |
| Scenario 5 | A | **3300** | 364 | 25,279 | 80 | 150 | 64,664,999 |
| | B | 3600 | 364 | 26,196 | 79 | 150 | **64,474,069** |
| | C | 3585 | **276** | 25,327 | 84 | **149** | 65,612,889 |
| | D | 3315 | 363 | 25,640 | 81 | 150 | 64,937,535 |
| | E | 3705 | **276** | **24,954** | 85 | 152 | 65,461,382 |
| | F | 3390 | 364 | 26,014 | **69** | 163 | 70,376,530 |

Ad. b). Weights:

The values of weight of each of six defined criteria were determined on the basis of responses to the conducted survey. The reason the same weights were applied to criteria C_03 and C_06 and also C_04 and C_05 was due to responders' actual misunderstanding of questions that had been stated in the poll. The reliability of the questionnaire survey can be leveraged by looking at the small number of responses received. Yet, since questions required clear answers, only highly ranking professionals with daily insight into modeling could have been accepted as relevant responders. Despite the target team being relatively small, there were professionals behind all feedback received. The goal set for this

article was solely to illustrate possibilities in how alternatives could be compared via the PROMETHEE method. The purpose was not, then, to deliver any set of unequivocal weights of criteria that in turn should be applied into the decision-making process. Questions in the questionnaire addressed very fine details, and already having received answers, we found out that responders had not really noticed these fine details we had asked them. Thus, in the case of the points mentioned above, they were clubbed, and the responses received are shared for these pairs (Table 5).

**Table 5.** Weights assigned to criteria.

| Criterion | C_01 | C_02 | C_03 | C_04 | C_05 | C_06 |
|---|---|---|---|---|---|---|
| Weight | 298 | 241 | 187 | 219 | 219 | 187 |
| Min/Max | Min | Min | Min | Min | Min | Min |

At this point, the question should be asked if these six criteria that had been predefined by the BPDRR committee proved to be sufficient to enable a correct evaluation of alternatives. According to the authors, there were numerous other factors that had to be applied to analyse a problem of such complexity. During discussions with responders, it turned out material availability in the warehouse, road conditions, issues related to occupation of traffic lanes, access to industrial hydrants and fire tanks, and the need to supply water to the largest industrial consumers appeared to be the factors that had the greatest degree in deciding the sequence of repairs to be undertaken.

Ad. c) Preference function:

Visual Promethee enables the preference function to be expressed in six different types: linear, usual, V-shape, U-shape, level, and Gaussian (Figure 12). This function is based on three thresholds: preference, incomparability, and equivalence. Definition of threshold values is the key input required to measure the strength of preferences. In effect, the preference function values range between 0 and 1. In the case we analyzed, the usual type of the preference function was defined for all six criteria.

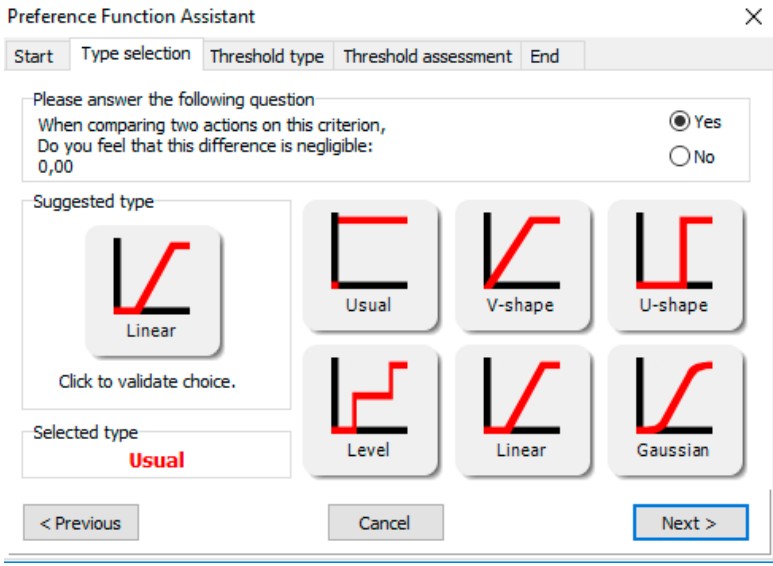

**Figure 12.** Promethee—definition of the preference function view.

The choice of the usual type implies that the decision maker would have strict preference for the alternative of the greatest value. Even in case of small differences among criterion values, the alternative with the higher value is selected [29].

Summing up the applied method, it should be highlighted that, rather than pointing out the 'right' decision, PROMETHEE and GAIA (Graphical Analysis for Interactive Aid) methods help decision

makers find the alternative that best suits their goals and their understanding of the problem [35]. It leads to a comprehensive understanding of the structure of the decision problem, identification and quantification of its conflicts and synergies, clusters of actions, and highlights the main alternatives and the supported structured reasoning [33].

## 3. Results

Within this project six types of rankings were defined for all pipes of the analysed water supply system, and objective functions were calculated according to criteria that had been provided. Based on the outcome from the previous step, a ranking of the actions for each of the scenarios is presented in Figure 13, Figure 14, Figure 15, Figure 16, Figure 17. Each of the figures listed below shows both the final outranking index as well as the net outranking flow.

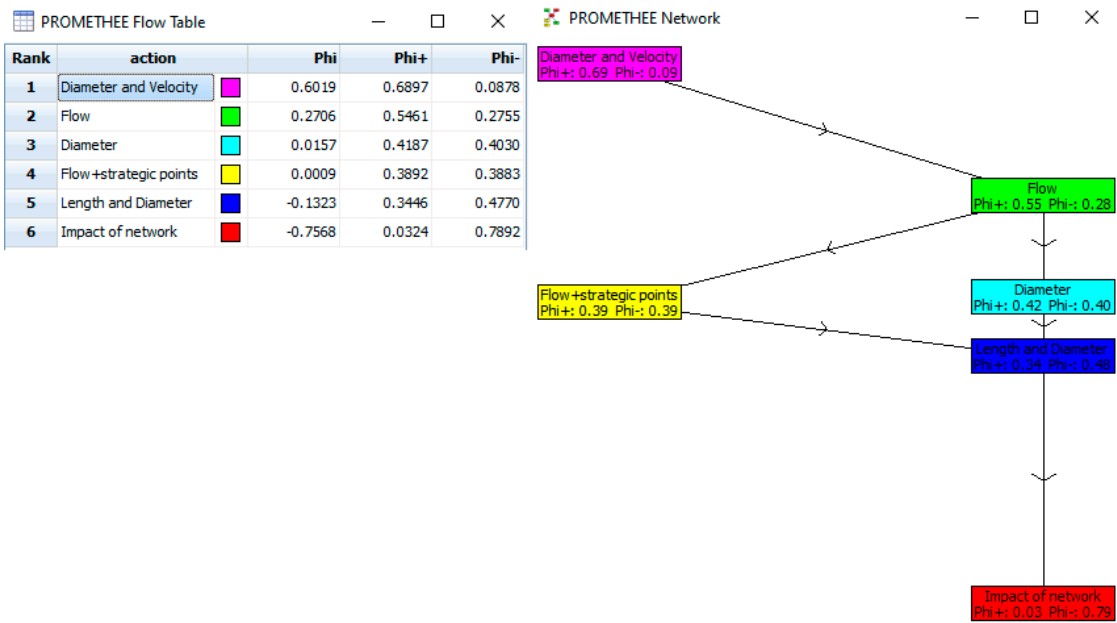

**Figure 13.** Final outranking index and net outranking flow of scenario 1.

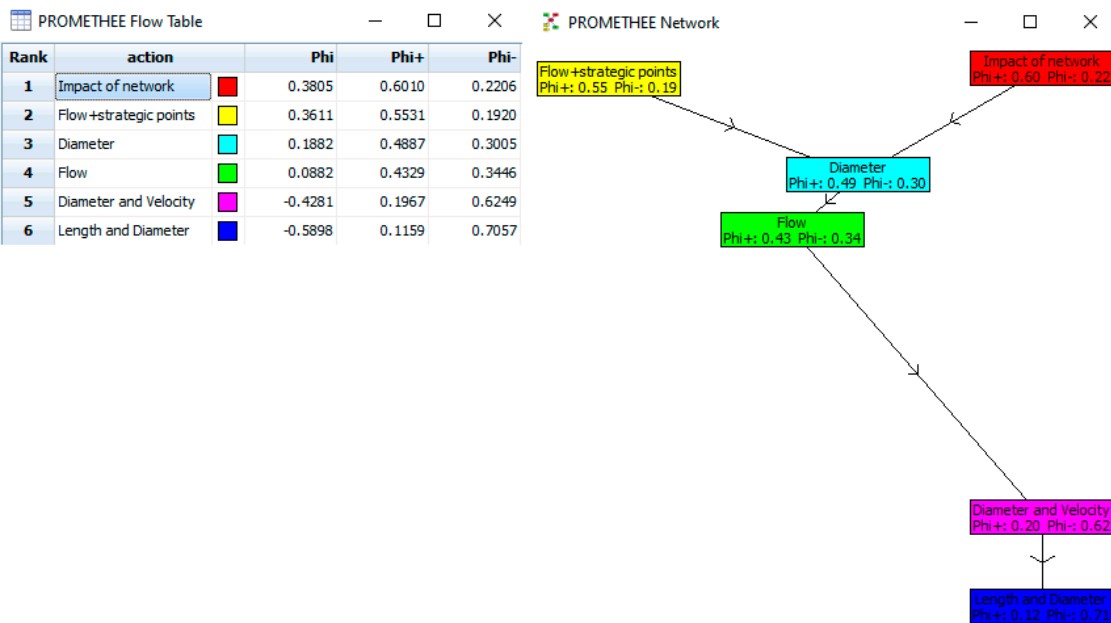

**Figure 14.** Final outranking index and net outranking flow of scenario 2.

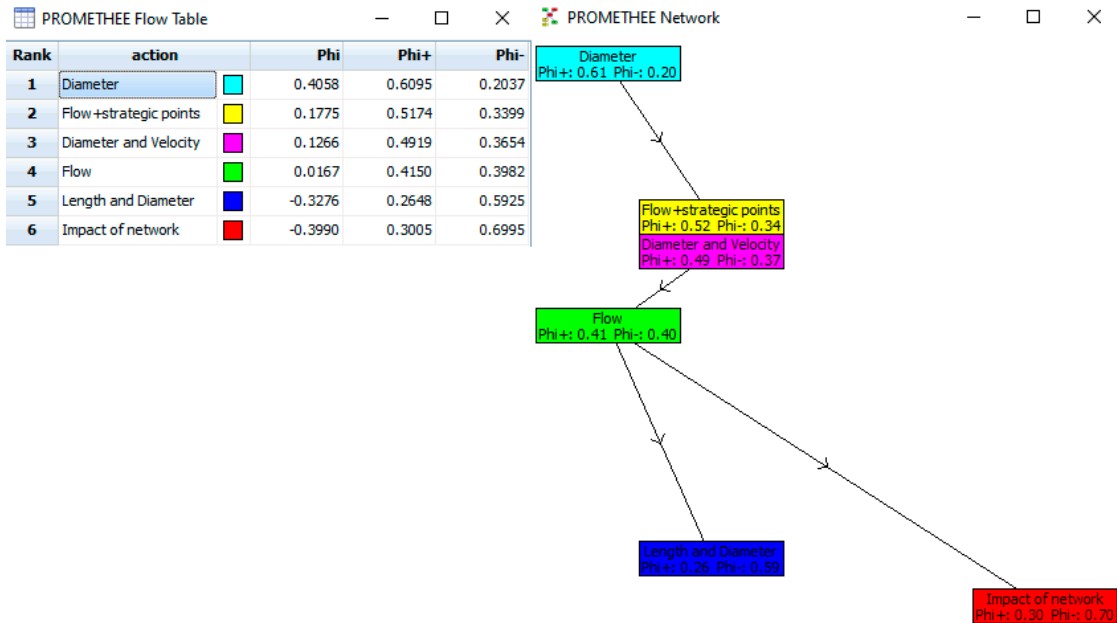

**Figure 15.** Final outranking index and net outranking flow of scenario 3.

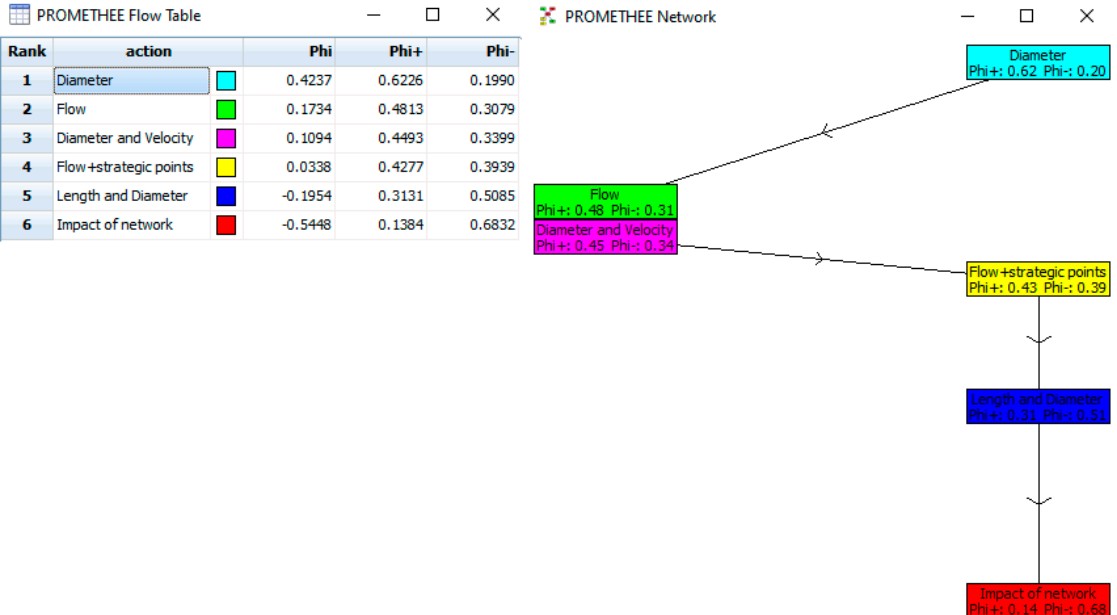

**Figure 16.** Final outranking index and net outranking flow of scenario 4.

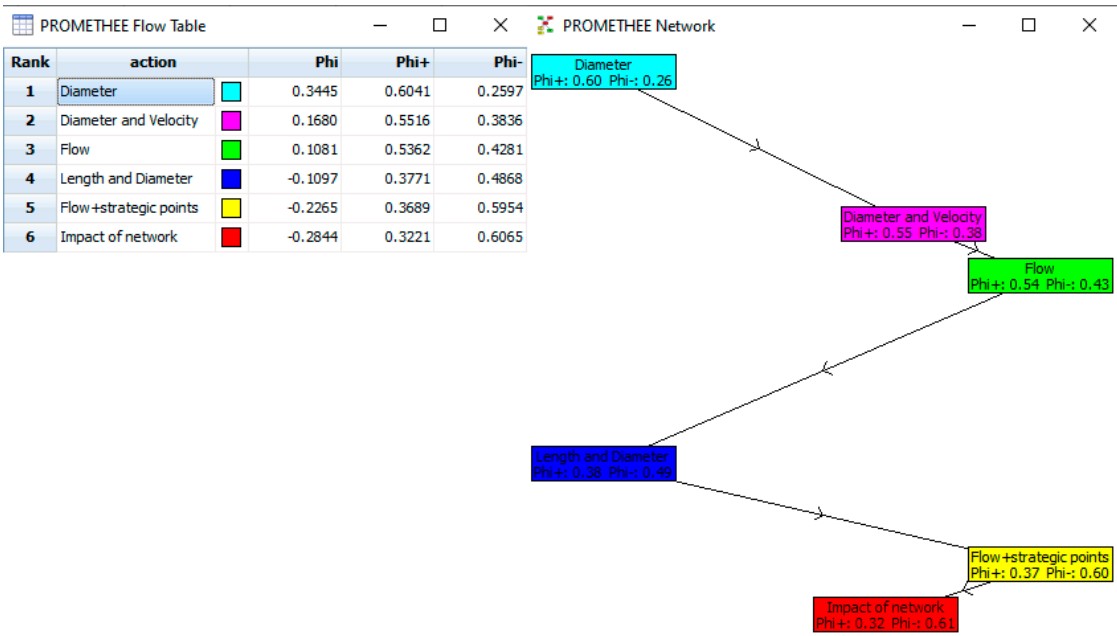

**Figure 17.** Final outranking index and net outranking flow of scenario 5.

Based on flow preference values (Phi) that have been presented in the PROMETHEE Flow Table, it is possible to answer the question which type of six rankings is the preferred one in this particular scenario. This makes it possible to draw the conclusion that both a change to a damage type and its localization in the network had a significant impact on the repair prioritization strategy.

This fact is one of many examples that proves why it is so difficult to define one dimension of operational resilience that can be expressed in one type of metric. Table 6 presents the best variants that have been obtained for the analysed scenario.

**Table 6.** Final results based on equations for the six criteria to evaluate the performance of a solution.

| Scenario | Best Ranking | Criteria | | | | | |
|---|---|---|---|---|---|---|---|
| | | C_01 | C_02 | C_03 | C_04 | C_05 | C_06 |
| | | (min) | (min) | (min%) | (min) | (-) | (L) |
| 1 | C (Diameter and Velocity) | 2970 | 6900 | 41,098 | 211.01 | 760 | 77,370,498 |
| 2 | F (Impact of network) | 3960 | 4140 | 18,608 | 37.45 | 120 | 59,438,741 |
| 3 | A (Diameter) | 3960 | 5460 | 26,224 | 36.5 | 73 | 82,986,155 |
| 4 | A (Diameter) | 2565 | 5460 | 32,555 | 79.71 | 166 | 74,506,054 |
| 5 | A (Diameter) | 3300 | 5460 | 25,279 | 79.82 | 150 | 64,664,998 |

Based on obtained results, it turns out that the definition of a single ranking of pipes is not possible when the selected hydraulic parameter, effective for all five selected scenarios, has to be met. As an example, while in four out of five analyzed scenarios the F ranking is the least preferred solution, it wins in scenario 2.

A sensitivity analysis was performed during the last stage of the multicriteria variant assessment. It enables the evaluation of the influence of the weight of each of the criteria on preferences of considered actions. An interesting property of the PROMETHEE methods is that preference flows are linear functions of the weights of criteria, and this feature makes it easier to perform sensitivity analyses. Details are presented in manual [36]. They can be carried out via one of a few available functionalities of Visual PROMETHEE (i.e., walking weight, visual stability intervals, and decision maker brain in GAIA plane). Visual stability intervals presented in scenario 1 for criterion 4 (Table 7 and Figure 18) show how the Phi multicriteria net flow scores change as the function of the weight of

the criterion. The horizontal dimension corresponds to the weight of the selected criterion, and the vertical dimension corresponds to the Phi net flow score.

**Table 7.** Stability range for scenario 1.

| Criterion | C_01 | C_02 | C_03 | C_04 | C_05 | C_06 |
|---|---|---|---|---|---|---|
| Weight granted | 22% | 18% | 14% | 16% | 16% | 14% |
| Stability interval (WSI) | 13.86–23.93% | 0–100% | 10.53–100% | 12.99–28.17% | 7.39–17.23% | 13.04–22.45% |

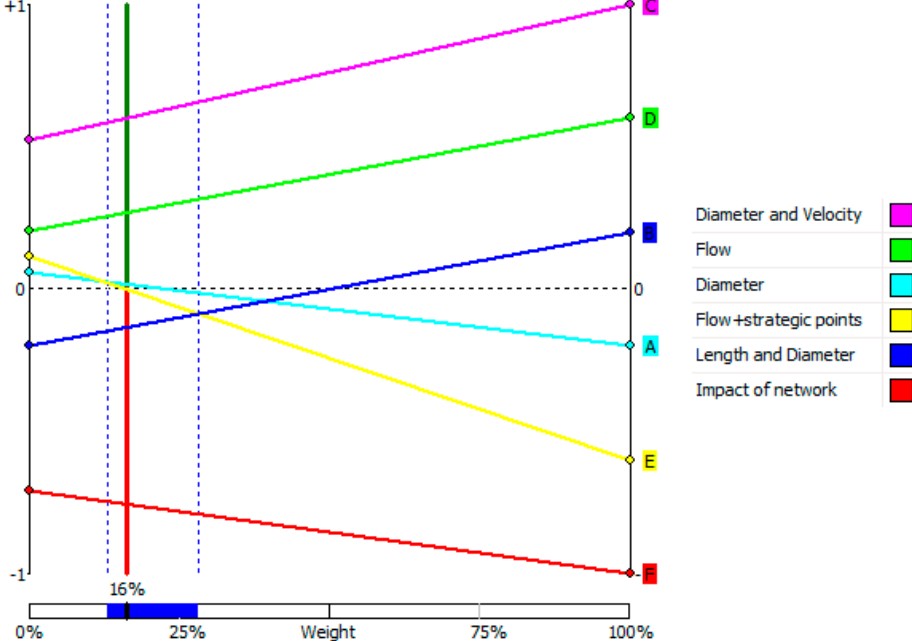

**Figure 18.** Visual stability interval for C_04 in scenario 1.

For each action, a line is drawn that shows the net flow score as a function of the weight of the criterion. On the right edge of the display the weight of the criterion equals 100%, and the actions are ranked according to that single criterion. On the left edge, the weight of the criterion equals 0%. The position of the vertical green and red bars corresponds to the current weight of the criterion. The intersection of the action lines with the vertical bar lets PROMETHEE II complete the ranking. The example presented above in Figure 18 can be taken as proof that the scores of ranking A (diameter) and ranking E (flow plus strategic point) go down when:

- the weight of criterion_04 (average time each node is without service) increases;
- while the score of ranking C (diameter and velocity), B (diameter and distance from the source), and ranking D (flow) increases.

Another form of the visualisation of Visual PROMETHEE results is the GAIA plane—a two or three dimensional representation of the multicriteria problem. Initially U, V and W dimensions are computed. U represents the maximum possible quantity of information, V provides the maximum additional information orthogonal to U, while W depicts the maximum additional information orthogonal to both U and V [36].

GAIA plane enables presentation of three different parameters at the same time:

- actions are represented by points;
- criteria are represented by vector;
- weights of the criteria are represented by red vector (called the decision axis).

Figure 19 presents the GAIA plane view developed for this decision problem. All GAIA planes for scenarios 1–5 (consecutive values of 82.8%, 91.3%, 76.3%, 95.4%, and 69.7%) where the percentage rank exceeds 60% are considered reliable and can be applied in the result interpretation. The red axis in Figure 19 is often called the decision axis (based on criteria weights). Criteria C_01, C_03, C_04, C_05, and C_06 whose axes share the same orientation with the decision axis, have stronger influences on the decision-making process, as opposed to C_02. The location of all alternatives A to F on the GAIA plane versus the decision axis reflects their preference of ranking (i.e., for scenario 1 alternatives: E (impact of network), F (flow + strategic points), and B (diameter and length) are located farthest from the direction axis (also shown in Figure 19)).

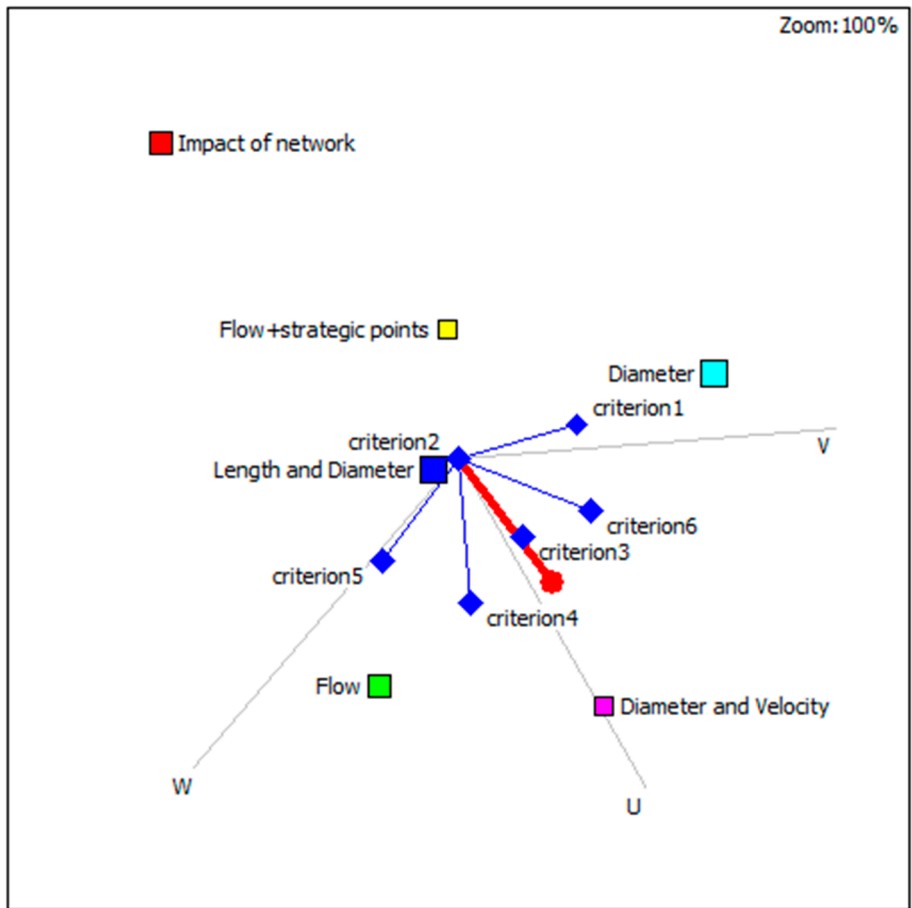

**Figure 19.** GAIA plane for scenario 1.

The length of the decision axis (actually the vector) is also an important factor. The longer this decision axis is at the GAIA plane, the stronger the decision power [35]. If otherwise, according to the weights, criteria strongly conflict and make it more than difficult to make any good decision on a compromising solution [37].

In our case, a long decision axis was found in all 5 scenarios. In the GAIA plane (Figure 19) for scenario 1, the diameter and velocity alternative was the closest to the decision axis and shared its orientation, which according to the PROMETHEE ranking, appoints it the best compromise solution.

The PROMETHEE analysis conducted in this study showed that for scenarios 3, 4, and 5, ranking A (based on diameter) was the most preferred solution. In the case of two others scenarios 1 and 2, ranking F (impact of pipe closures on the network's hydraulics) in parallel with ranking E (flow and pipes that lead to strategic points like hospitals and fire-flow nodes) and ranking C (diameter and velocity) were these of the best choice.

It is interesting why diameter, as a hydraulic parameter, is more significant than flow. It seems that in the case of cascading failures, the network's capacity is maintained even when flow paths heavily change because of rapid segment isolations, leakages, and pipe bursts. Moreover, in scenario 2, ranking F (based on flow and location of strategic points) turned out to be the most preferred one. The root cause lies in the fact that many damages in this scenario were located close to hospitals and fire flow nodes. This led us to the conclusion that determining the rank of critical pipes for scheduling repairs, with the purpose to obtain the best factors for the defined objectives function (criteria), should not include only factors based on hydraulic parameters and graph theory.

This proposed methodology allowed us to answer the question from the abstract of this paper. It appeared to be possible to obtain results in this particular case for each of five damage scenarios and six different criteria and compare them with five types of alternatives (defined ranking of pipes which would determine the scheduling of repairs).

The GAIA analysis equips decision makers with an additional perspective to look at multicriteria problems. It is much easier to understand the relation between criteria and alternative versus decision axis (which represents our best preference solution).

## 4. Conclusions

The aim of this article was to develop a method to schedule repairs in water distribution systems in terms of post-disaster response and restoration services. This article is a conference paper based on the preprint presented at the WDSA/CCWI Conference. The novelty in this article compared with the BPDRR research is a detailed presentation on developing the ranking methodology and evaluating the possible solution criteria and the impact of their weights. These have been made possible due to a PROMETHEE analysis, including GAIA plane and sensitivity analyses based on stability ranges, as an example.

Further analyses could be conducted with due consideration of other criteria, such as: bacteriological threats (water contamination), which is very important in networks with water quality problems (pipeline damage may let hazardous substances leak into the supply); weather conditions; tools and spares availability; road conditions in the area of the damage (cost of renovation); reserve water tanks for strategic points (i.e., hospitals); the possibility to connect to different networks or temporary supplies from different ends; fire assessment, the threat to people, and the distance to real operational hydrants; industrial plants in the area affected by damages; distance between damaged spots from the perspective of maintenance crew access (time of arrivals); power planning for strategic points (i.e., hospitals); number of people at risk in strategic points; and the need to close supply for large areas to reduce the risk of flood caused by outflow of large amounts of water. Future investigations should consider the development of an algorithm that could be integrated with the model, GIS, SCADA (Supervisory Control And Data Acquisition), and possibly other systems running on data that have been mentioned above.

Based on obtained results, a conclusion can be made that the methodology related to the planning schedule of the removal of damages should be dynamic. The restoration or repair of subsequent sections of the network affects hydraulic conditions in the water supply system, and it should take into account the weight of each criterion depending on where the failure occurred.

Our motivation behind the proposed approach was the desire to propose a robust, fully reliable solution as a guide for decision makers and operators. The method we strived to develop may be incorporated into the core engine of emergency restoration programs and can be made available to water utilities staff, whose key principle is to provide a reliable solution to their crews in as short period of time as possible. Furthermore, the standards currently applied into water safety plans enforce the necessity to define critical locations into every water distribution system (WDS).

The ranking approach to scheduling repairs of WDS in the Post Disaster Response and Restoration Service that has been presented in this paper enables an accurate sequencing of repair works on damages caused by an earthquake to be determined. The proposed framework includes both expert

knowledge to identify the minimum number of pipe segments required to isolate as well as the MATLAB Toolkit algorithm intended to generate the sequence of tasks and assign them to individual repair teams. The suggested method makes it possible to evaluate various solutions by the chosen decision-making support software.

An interesting task to be realized in the future would be to conduct more detailed research on the impact of additional criteria, not analyzed in this article yet listed in the second paragraph of this summary, on the sequence of undertaken repairs.

Summing up, work on this article was an inspiration for the entire team to undertake a far more detailed questionnaire and research its results, aggregated by threat type, which could be further associated in the decision-making process with selected criteria.

**Supplementary Materials:** The following are available online at http://www.mdpi.com/2073-4441/11/8/1591/s1, Files '104_Solution_Template.xlsx' and 'Damage_Scenarios.xlsx' have been archived as the Suplementary.rar file.

**Author Contributions:** Conceptualization, A.B., R.B., J.B. and P.Z.; methodology, A.B., R.B., J.B. and P.Z.; software, R.B. and P.Z.; validation, A.B. and J.B.; formal analysis, A.B. and J.B.; writing—original draft preparation, A.B., J.B., R.B. and P.Z.

**Funding:** This research was funded by Ministry of Science and Higher Education. Research subsidy number: 504101 09/91/SBAD/0678.

**Acknowledgments:** Delivery of this article was possible thanks to the free access to the following software: Epanet, Epanet-Matlab Toolkit and Visual Promethee Academic Version.

**Conflicts of Interest:** The authors declare no conflict of interest.

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
