# Peer review of "Ranking Approach to Scheduling Repairs of a Water Distribution System for the Post-Disaster Response and Restoration Service"

_water, doi:10.3390/w11081591_

Round 1
Reviewer 1 Report
This study developed the Ranking Approach to Scheduling Repairs of Water Distribution System in Post Disaster Response and Restoration Service Curtain Weirs for algal blooms control in a typical tributary.
The topic of this study is within the scope of the Journal. However, some explanations and errors make it difficult to understand the contents of this study. Therefore, the reviewer would like to defer the decision until the authors revise the paper based on the comments below. The authors should address each of the comments successfully for possible publication.
Major
1. Is the improvement of the solution only related with the evaluation of the delivered flow to the users?
2. This study considered PDA approach for hydraulic analysis and to analysis PDA various kinds of HOR (Head Outflow Relationship) can choose. So, The authors should explain more detail about the necessity of PDA for this study and the characteristic of each HOR function.
3. To evaluate the effective improvement, it is necessary to furnish clear indication.
4. For the application network, is there some results in terms of sensitivity analysis when weight coefficients change?
5. Do you build a software? If so, what is the programming language and how long it takes to run all the simulations? For the readers, it is important to know if the solution is too time-consuming.
6. About the objective function, did you apply the optimization technique? Generally the terms of objective function in WDSs field is used for optimization. If the authors applied optimization approach, this study needs to declare and adds the supplement explanation about the optimization technique (e.g., decision variables, applied their own algorithm parameter values)
7. The authors did not evaluate the cost, which is a relevant cost in this paper and could affect the results.
8. In the case of a questionnaire survey, if you get the high reliability of questionnaire survey result, the number of specimens should be high. However, this study only 46 specimens applied for evaluating the weight even though they are high-quality specimens such as the top management employees from water utilities.
9. To improve the reliability of results for the questionnaire survey, the reviewer recommends increasing the number of specimens for the survey.
Author Response
Response to Reviewer 1 Comments
As attachment I sent also two supplementary files, hoping to better understand our work which results are not included into manuscript.
Major
1. Is the improvement of the solution only related with the evaluation of the delivered flow to the users?
Response 1:
The limitations in undertaken approach are the six criteria (listed and described in lines 81-89) that were dictated by organizers of Battle of Water. Most of these criteria are related to maintenance of required flow and pressure via such parameters: C_05 – Number of nodes that awaited servicing for more than 8 consecutive hours, C_06 – volume of water lost over seven days that followed the event or C_04 – average length of time when nodes were out of service, in minutes.
The main undertaken challenge is the planning of repair activities. The key here is to align to aspects, first being the focus on system maintenance in best available condition and return to 100% efficiency. While the second one is to maintain waterflow through the vast majority of nodes, including critical points, such as hospitals or sites currently on fire.
Obviously, criteria we’ve listed do not fully exploit all aspects of the system. They should be regarded as limitations we’ve applied to realize the BPDRR research. Possible directions of further research have been suggested in lines (499-533).
2. This study considered PDA approach for hydraulic analysis and to analysis PDA various kinds of HOR (Head Outflow Relationship) can choose. So, The authors should explain more detail about the necessity of PDA for this study and the characteristic of each HOR function.
Response 2:
Following suggestions of the reviewer, text has been updated in lines (147-162)
During the first hours of the simulation, in some nodes there are firefighting demands – 35?/? (please check ‘Damage_Scenarios.xlsx’ attached. As dictated, each site on fire is extinguished with precisely 756?3 of water. However, since pressure deficient conditions are met in the network during most of the restoration time, the demand cannot be fully supplied. To model that partial supply, pressure driven analysis (PDA) must be used. Wagner’s power relationship is built-in the proposed method.
(1) |
In order to realize the algorithm where pressure-driven demands (FCV, TCV, CV, Reservoir) have been incorporated, additional four artificial elements would be required for all model’s nodes. Further details on PDD calculation in Epanet can be found in the following articles: Paez D., Suribabu C., Filion Y: https://ojs.library.queensu.ca/index.php/wdsa-ccw/article/view/12024
During calculations we based on five copies of EPANET2 model of the WDN under pre-disaster conditions and five more with the location and configuration of the damages from “Damage_Scenarios.xlsx”.
3. To evaluate the effective improvement, it is necessary to furnish clear indication.
Response 3:
The aim of our article isn’t any improvement of the way failures are repaired but verification if there are clear and objectively measureable criteria that defined the sequence of tasks undertaken. For five different scenarios of breakdowns it has been proved that influence of locallisation and failure type have seriously changes the way actions are undertaken (table 6). It has been clearly indicated in abstracts (18-20, 61-64). Following suggestions of the reviewer, text has been updated in lines (Conclusion).
4. For the application network, is there some results in terms of sensitivity analysis when weight coefficients change?
Response 4:
Yes. In the article, at Fig.18 and in the description, in lines (421-423 and 441-452) it has been presented how a change of the weight of a criterion implies the change on preferred rankings. Unfortunately, the Academic licence of Visual Promethee, used during our work on this article did not allow any simultaneous comparison of a few criteria, at one graph and in relations to a single criterion. As soon as the final report of BPDRR has been published we find it highly interesting to compare the influence of weights, as indicated by 9 research teams, on achieved sequence of undertaken repairs, and all these for all five scenarios of events.
5. Do you build a software? If so, what is the programming language and how long it takes to run all the simulations? For the readers, it is important to know if the solution is too time-consuming.
Response 5:
Delivery of no new software has been required for the needs of the realisation of this task. Simulations have been conducted using the Matlab Environment. Epanet’s model calculation has been executed directly in Matlab with the Epanet-Matlab toolkit installed (line 259-264). This package utilizes the Epanet.dll library for hydraulic calculations what enables further processing of objective functions. Setup of controls within the hydraulic model was required for the modelling of the maintenance of brakes and leaks (isolation, reparation and replacement). A dedicated Matlab script has been prepared solely for that purpose. Calculation of the objectives function for 6 criteria requires around 1h. It is worth mentioning that the complete calculation process has been realized at a standard PC (Intel i5, 16Gb) of average processing speed. Indeed, we do agree in this very case the runtime of calculations performed was not the time limit. If any real deployment were to be found for solutions of this type, it would be required to specify the time, necessary for an operator, to make a decision with use of our solution as the decision-supporting system. (264-271).
6. About the objective function, did you apply the optimization technique? Generally the terms of objective function in WDSs field is used for optimization. If the authors applied optimization approach, this study needs to declare and adds the supplement explanation about the optimization technique (e.g., decision variables, applied their own algorithm parameter values).
Response 6:
To deliver this article Promethee has been used as one of multicriteria optimisation methods, yeti t has been apllied only to compare six various types of rankings (approaches) rather than explicite. However, table 4 has been updated and an additional clarification (338-344) added, to follow up upon reviewer’s request.
7. The authors did not evaluate the cost, which is a relevant cost in this paper and could affect the results.
Response 7:
This is true. The cost of repairs has not been any of the criteria that impacted the choice of pipes to be repaired. Since the research was conducted in terms of BPDRR, no additional criteria could have been added upon these officially dictated. This is, however, an interesting aspect and it has been included in the summary (lines 497-499).
8. In the case of a questionnaire survey, if you get the high reliability of questionnaire survey result, the number of specimens should be high. However, this study only 46 specimens applied for evaluating the weight even though they are high-quality specimens such as the top management employees from water utilities.
9. To improve the reliability of results for the questionnaire survey, the reviewer recommends increasing the number of specimens for the survey.
Response 8,9:
The goal set for this article has been solely to illustrate possibilities how alternatives could be compared via the Promethee method, the purpose wasn’t then to deliver any set of unequivocal weights of criteria that in turn should be applied into the decisionmaking process.
Indeed, reliability of the questionnaire survey can be leveraged by looking at numbers. Yet please note it was realized with a short time – BPDRR– and only high rank professionals with daily insight into modeling could have been accepted as relevant responders. Depite the target team is relatively small, there are professionals behind all feedback received (lines 351-353).

Reviewer 2 Report
The manuscript has the purpose to apply a ranking approach for water distribution system repaire scheduling. The topic of this study makes sense, but there are some main criticisms. Please see comments listed below.
1) Compared with the previous paper by the authors (battle of post-disaster response and restauration (BPDRR), presented in SDSA/CCWI 2018), I can't find novelty of this study. The overall contents of the manuscript, analysis method and results are the same as the previous study, and even the tables and figures are the same. Therefore, please clarify the originality of the paper compared with the existing research. 2) Please revise the figures in the manuscript for better understanding. (e.g. Figure 2) In addition, please check the numbers of figure and table. 3) Line 146) The authors mentioned, the EPANET using the PDA method was used in the study. However, as far as I know, the PDA is not included in the EPANET 2.0 (the most recently officially released version). If you applied other model, you need to mention the exact name of the model. Because, there are various PDA models based on the EPANET, and the results can vary depending on which PDA model is used. 4) Line 205) In this study, the distance between pipe and the source is considered to to calculate the ranking. However, I think it is reasonable to consider the actual flow length of water rather than the straight-line distance. In addition, I also think that for a system with more than one source, you need to consider both the flow rate from each source and the flow length at the same time. For this you can use EPANET's source tracing tool. 5) Line 230) In the analysis results, there are some negative pressure nodes. However, you should check whether this negative pressure is due to the isolation of a particular area, or simply because of the pressure drop. If the negative pressure is due to the isolation of a particular area, the reliability of the analysis results for the entire system will be reduced. In addition, in order to clearly identify the negative pressure nodes, please add a condition below 0m in the legends of figures 9 and 10. 6) Table 3) I think the number of valves to isolate pipe-437 should be 2. 7) Please check references again. For example, in [12], the name of the author is mislabeled (the first and last names are reversed); in [23], the link information can not be verified.
Author Response
1) Compared with the previous paper by the authors (battle of post-disaster response and restoration (BPDRR), presented in SDSA/CCWI 2018), I can't find novelty of this study. The overall contents of the manuscript, analysis method and results are the same as the previous study, and even the tables and figures are the same. Therefore, please clarify the originality of the paper compared with the existing research.
Response 1:
Our article is an extension of our proposal that was presented at WDSA 2018 conference as ‘FinalPaper104’. The solution depicted at 8 pages had a purpose of bringing that complicated problem closer to readers who may have not had a chance to join the WDSA/CCWI 2018 conference. Reviewers’ responses published earlier were mainly focused on a need to improve English, document’s structure. In today’s shape this paper is a lot more complete than the conference document. We do hope this expanded contents will be positively accepted by reviewers.
2) Please revise the figures in the manuscript for better understanding. (e.g. Figure 2) In addition, please check the numbers of figure and table.
Response 2:
Figure 2 has been updated. Additional figures in sub-chapters 2.1, 2,2 have also been corrected.
3) Line 146 The authors mentioned, the EPANET using the PDA method was used in the study. However, as far as I know, the PDA is not included in the EPANET 2.0 (the most recently officially released version). If you applied other model, you need to mention the exact name of the model. Because, there are various PDA models based on the EPANET, and the results can vary depending on which PDA model is used.
Response 3:
During the first hours, in some nodes there are firefighting demands – please check attachments ‘Damage_Scenarios.xlsx’ of 35?/?. As dictated, each is extinguished with precisely 756?3 of water. However, since pressure deficient conditions are met in the network during most of the restoration time, the demand cannot be fully supplied. To model that partial supply, pressure driven analysis (PDA) must be used. Wagner’s power relationship (1) is built-in the proposed method.
(1) |
In order to realize the algorithm where pressure-driven demands (FCV, TCV, CV, Reservoir) have been incorporated, additional four artificial elements would be required for all model’s nodes. Further details how PDD calculation in Epanet was described in detail in the following articles: Paez D., Suribabu C., Filion Y: https://ojs.library.queensu.ca/index.php/wdsa-ccw/article/view/12024
We based on five copies of EPANET2 model of the WDN under pre-disaster conditions and five more with with the location and configuration of the damages from “Damage_Scenarios.xlsx”.
Following suggestions of the reviewer, text has been updated in lines (lines 147 - 162).
4) Line 205: In this study, the distance between pipe and the source is considered to calculate the ranking. However, I think it is reasonable to consider the actual flow length of water rather than the straight-line distance. In addition, I also think that for a system with more than one source, you need to consider both the flow rate from each source and the flow length at the same time. For this you can use EPANET's source tracing tool.
Response 4:
Indeed, ranking that’s based on source tracing from a given node to the source should be tested. Our intention behind choosing this ranking was to specify the distance from a damaged pipe to the key water pumps facility. The tracing tool cannot be used to calculate the distance directly from a node to the source. It has already been planned to include time water needs to flow from any given source into the future research paper. It will be interesting also to include time required to reach specific sites of failure, based on a GIS map – that wasn’t provided within BPDRR.
5) Line 230) In the analysis results, there are some negative pressure nodes. However, you should check whether this negative pressure is due to the isolation of a particular area, or simply because of the pressure drop. If the negative pressure is due to the isolation of a particular area, the reliability of the analysis results for the entire system will be reduced. In addition, in order to clearly identify the negative pressure nodes, please add a condition below 0m in the legends of figures 9 and 10.
Response 5:
During analises we’ve had certainty that any reduction in pressure in this area is the effect of certain pipelines closed. According to model’s results in regular conditions (pipe not closed) system operates properly. In the normal conditions model there is no negative pressure for any time step. Negative pressures, however, can be observed in the model only after selected pipes have been closed. This has been clearly indicated in updated line (lines 249-250). During normal system exploitation water supply to every node and in all time steps has been met. Thus a closure of a single pipe ends up with decreased pressure.
Indeed, in case of the pressure value, analysis credibility is reduced. Yet the goal of the analysis is only to verify the influence of pipe’s closing on a number of consumers that will be out of service. With that goal in mind, credible results of calculations are not required, it is sufficient to highlight out of service nodes to which water has not been delivered. This has now been clearly mentioned in line (250-252): In course of analysis for certain nodes negative pressure has been received. This is interpreted as a lack of water supply to consumers that have been assigned specific nodes. Consumers with no access to water supply (calculated as the sum of water usage in out of service nodes) are the measure of the influence of a pipe’s closure on network’s hydraulics.
Additionally, figures 9 and 10 have been updated according to the recommendation.
6) Table 3) I think the number of valves to isolate pipe-437 should be 2.
Response 6:
Indeed, there has been a typo. Pipe 6069 has one valve and not zero. Table has been then updated by 6069 (1), while the number of valves to be closed remains unchanged.
7) Please check references again. For example, in [12], the name of the author is mislabeled (the first and last names are reversed); in [23], the link information cannot be verified.
Response 7:
References have been updated, following this suggestion. All other points in bibliography have also been reviewed.

Reviewer 3 Report
Overall, the paper explains in great detail the background, methodology, and results of the ranking approach to schedule repairs and replacements in case of an earthquake. I have just a few minor comments that I outlined below, by line number. Also, as a general comment, please recheck the English grammar and spelling of the manuscript; there are some improvements needed there. Good paper overall!
Minor corrections in rows 101 and 102, instead of "analyzes" it should be "analysis" and "analyses", respectively (please correct it in the remained of the manuscript).
Line 245: "breaks". not "brakes"
Line 251: How many valves belong to a pipe
Line 254: in accordance "with" the selected ranking
Lines 324-326: Please give examples on what other numerous factors the authors consider should be applied?
It is explained in the conclusions, but it would be nice to just briefly mention (list) a few examples here as well.
Lines 390-391: Please combine the last two sentences: "The horizontal dimension corresponds to the weight of the selected criterion, and the vertical dimension corresponds to the Phi net flow score."
Lines 456-489: General comment to possibly add in the conclusions section. Can you address some of the limitations of the study/method? You mentioned that some other criteria could be considered. Are there any other consideration, such as expert panel changes for example?
General comment/question: Since expert opinion is such a subjective matter, and I am not particularly familiar with the PROMETHEE software, what are some of the consistency measures determined to ensure that there is consensus between experts?
Author Response
Overall, the paper explains in great detail the background, methodology, and results of the ranking approach to schedule repairs and replacements in case of an earthquake. I have just a few minor comments that I outlined below, by line number. Also, as a general comment, please recheck the English grammar and spelling of the manuscript; there are some improvements needed there. Good paper overall!
Point 1: Minor corrections in rows 101 and 102, instead of "analyzes" it should be "analysis" and "analyses", respectively (please correct it in the remained of the manuscript).
Response 1:
Typo error, corrected.
Point 2: Line 245: "breaks". not "brakes"
Response 2:
Typo error, corrected.
Point 3: Line 251: How many valves belong to a pipe
Response 3:
Typo error, corrected.
Point 4: Line 254: in accordance "with" the selected ranking
Response 4:
Typo error, corrected.
Point 5: Lines 324-326: Please give examples on what other numerous factors the authors consider should be applied?
Response 5:
Text updated with the following lines 366-369: ‘During discussions with respondents it turned out that materials’ availability in the warehouse, road conditions, issues related to an occupation of a traffic lane, access to industrial hydrants and fire tanks, need to supply water to the greatest industrial consumer have appeared to be these factors that in greatest degree decided on the sequence of repairs undertaken’.
It is explained in the conclusions, but it would be nice to just briefly mention (list) a few examples here as well.
Point 6: Lines 390-391: Please combine the last two sentences: "The horizontal dimension corresponds to the weight of the selected criterion, and the vertical dimension corresponds to the Phi net flow score."
Response 6:
Text updated in line 429-430.
Point 7: Lines 456-489: General comment to possibly add in the conclusions section. Can you address some of the limitations of the study/method? You mentioned that some other criteria could be considered. Are there any other consideration, such as expert panel changes for example?
Response 7:
Text updated: (line 497-499): Because the article is an extension of the publication presented at the WDSA/CCWI Conference, the authors are aware of the limitations that have been made during the implementation of the Battle of Models study
Point 8: General comment/question: Since expert opinion is such a subjective matter, and I am not particularly familiar with the PROMETHEE software, what are some of the consistency measures determined to ensure that there is consensus between experts?
Response 8:
The following text has been added (528-533) : ‘An interesting task to be realized in the future would be to conduct a more detailed research on an impact of additional criteria, not analysed in this article yet listed in the second paragraph of this summary, on the sequence of undertaken repairs.
Summing up, work on this article was an inspiration for all the team to undertake far more detailed questionnaire and research that’s results, aggregated by threat type, could be further associated in the decisionmaking process with selected criteria’.

Round 2
Reviewer 1 Report
The authors completed overall my comments.
But I think, this paper almost similar to BPDRR paper which presented in WDSA 2018.
Although the authors used the same topic and concept as BPDRR paper, the contents in this manuscript should be changed.
Therefore, I strongly recommand clarify the novelty of this article compared with the BPDRR research.
Author Response
Response to Reviewer 1 Comments
(Round 2)
Comments and Suggestions for Authors.
The authors completed overall my comments.
But I think, this paper almost similar to BPDRR paper which presented in WDSA 2018.
Although the authors used the same topic and concept as BPDRR paper, the contents in this manuscript should be changed.
Therefore, I strongly recommand clarify the novelty of this article compared with the BPDRR research.
Date of this review
20 Jul 2019 07:06:32
22 July 2019
Dear Reviewer,
First of all – thank you for your positive feedback. Your comments are highly valuable. Responding to your doubt of the content of the delta between the BPDRR paper and the manuscript we’ve submitted, please find here the key novelties that have been included:
· Truly, results we’d obtained that have been discussed in both papers are based on the same table (Table 6) with results. Though, there are novelties in the manuscript. The Promethee method has been described from the scratch, with step-by-step instructions, followed with an analysis of the GAIA graph and the visualization of its results. Furthermore, a sample of the Sensitivity Analysis has enabled a more complex review of the considered decision-making problem. All points mentioned above, here described in detail, were mentioned in BPDRR in just this single sentence ‘Based on Visual Promethee software and solution template file, final results for each scenario were obtained (Table 2)’.
· Please note that while BPDRR should be processed as a Preprint (in line with clarifications published by MDPI[1]), this manuscript should be regarded as a high quality conference paper.
According to the comment of the reviewer, the following content has been changed in the manuscript (first paragraph of conclusion chapter).
“The aim of this article has been to develop the method to schedule repairs of a Water Distribution System in Post Disaster Response and Restoration Service. This article is a conference paper based on the preprint presented at the WDSA/CCWI Conference. The authors are aware of the limitations that must have been made during the implementation of the Battle of Models study. Certainly, the novelty in this article compared with the BPDRR research is the presentation of the evaluation of possible solution criteria and the impact of their weights. These have been made possible due to the PROMETHEE analysis: GAIA plane and sensitivity analysis based on stability ranges, as an example”.
Yours sincerely,
Alicja Bałut
[1] https://www.mdpi.com/journal/psych/instructions

Reviewer 2 Report
I think the authors have modified the paper to reflect the opinions of the reviewers. There are still some assumptions and limitations in the paper, but this is explained in the paper. I hope that these limitations will be complemented in future studies.
However, the figures included in the manuscript are still unreadable. In particular, the meaning of Figures 3 and 4 is difficult to understand and seems unnecessary. Authors should improve their readability by increasing the resolution and font size of inserted figures overall.
In addition, the information on the authors of [12] should be changed to 'Yoo D.G., Jung D., Kang D., Kim J.H.'.
Author Response
Response to Reviewer 2 Comments
(Round 2)
Comments and Suggestions for Authors
I think the authors have modified the paper to reflect the opinions of the reviewers. There are still some assumptions and limitations in the paper, but this is explained in the paper. I hope that these limitations will be complemented in future studies.
However, the figures included in the manuscript are still unreadable. In particular, the meaning of Figures 3 and 4 is difficult to understand and seems unnecessary. Authors should improve their readability by increasing the resolution and font size of inserted figures overall. In addition, the information on the authors of [12] should be changed to 'Yoo D.G., Jung D., Kang D., Kim J.H.'
Submission Date
28 June 2019
Date of this review
19 Jul 2019 07:46:12
22 July 2019
Dear Reviewer,
First of all, thank you for the positive review of my article. I do appreciate all your feedback.
Your comment concerning unreadability of Figures 3 and 4 – point taken. I have replaced them with new ones where all differences have been distinctively marked.
And referring to my mistake in authors’ names, in point 12 of bibliography – sincere apologies. I have provided them now exactly in the form you’ve suggested.
Yours sincerely,
Alicja Balut
